# Phages in Anaerobic Systems

**DOI:** 10.3390/v12101091

**Published:** 2020-09-26

**Authors:** Santiago Hernández, Martha J. Vives

**Affiliations:** 1Department of Biological Sciences, Universidad de los Andes, Bogotá 111711, Colombia; s.hernandez41@uniandes.edu.co; 2School of Sciences, Universidad de los Andes, Bogotá 111711, Colombia

**Keywords:** anaerobic, aerobic, facultative, bacteriophage, bacteria

## Abstract

Since the discovery of phages in 1915, these viruses have been studied mostly in aerobic systems, or without considering the availability of oxygen as a variable that may affect the interaction between the virus and its host. However, with such great abundance of anaerobic environments on the planet, the effect that a lack of oxygen can have on the phage-bacteria relationship is an important consideration. There are few studies on obligate anaerobes that investigate the role of anoxia in causing infection. In the case of facultative anaerobes, it is a well-known fact that their shifting from an aerobic environment to an anaerobic one involves metabolic changes in the bacteria. As the phage infection process depends on the metabolic state of the host bacteria, these changes are also expected to affect the phage infection cycle. This review summarizes the available information on phages active on facultative and obligate anaerobes and discusses how anaerobiosis can be an important parameter in phage infection, especially among facultative anaerobes.

## 1. Introduction

Phages, the most abundant biological organisms on Earth, regulate bacterial populations by means of their lytic activity. Bacteria actively participate in biogeochemical cycles [1], especially in the carbon and nitrogen cycles [2,3,4]; also, they produce a great percentage of the atmospheric oxygen (cyanobacteria), and recycle the oxygen in the soil [5]. Thus, phages influence nutrient flow and the metabolic processes in nature in which bacteria are involved, a significant proportion of which occur in anaerobic environments, such as sediments and deep seas. This means that the ecological implications of bacterial metabolism in the absence of oxygen are of great importance in balancing the different biogeochemical cycles. In complex animals, prokaryotic cells—most of which are found in the intestines and play a fundamental role in nutrition and defense—are estimated to outnumber eukaryotic cells. Bearing in mind that animals’ intestines provide an anoxic environment, bacterial anaerobic metabolism is essential in the homeostasis of macro-organisms [6,7]. Animal intestines are also subject to the action of phages and there is increasing evidence of their importance in the regulation of intestinal microbiota [8,9].

As intracellular bacterial pathogens, phages are dependent on their host for replication and their successful infection depends on the metabolic state of the bacteria. In facultative anaerobes, the transition from an aerobic to an anaerobic environment involves modifications in the metabolic pathways that are activated or inactivated, and that can, in turn, generate changes in the bacteria’s physiology. These metabolic and physiological changes in the bacteria are likely to affect infection and viral replication. However, very few studies have been conducted on phages that infect anaerobic bacteria, and little is known about the behavior of anaerobic phages when infection occurs under anoxic conditions.

This review compiles the existing state of the art on strict and facultative anaerobic phages, infection processes in these bacteria, and the known effects that the absence of oxygen can have on infection efficiency and viral replication.

## 2. Bacteriophage Diversity

Since the beginning of the 19th century, phages have been studied as both biological entities and as molecular models. The period between the end of the 1930s and the beginning of the 1940s was the most important for research on these viruses [10]. In order to understand the infection process of bacterial viruses, Ellis and Delbruck proposed an experimental design, known as a one-step or single-step growth curve, to evaluate the infection cycle of phages [11]. In 1959, Adams published *Bacteriophage*, which soon became the world’s first reference book for the study of these organisms. Although there was a growing body of information on the biology and diversity of phages, there was no established taxonomy for these viruses [12]. As a result, in 1987, Ackerman proposed the first taxonomic model based on microscopic images [13].

In the field of molecular biology, bacteriophages were of great importance in the experiments conducted by Benzer in 1955 using bacteriophage T4, through which the molecular concept of the gene was discovered [10]. The study of the phages also made it possible to identify the transduction process, decipher the genetic code, and discover the messenger RNA [12].

A wide variety of phages are known to exist, and based on isolation and visualization techniques developed, these have been characterized phenotypically according to lysis plate morphology, host range, and virion morphology [14]. On the same basis and with increasing emphasis, phages have been grouped according to their type of genetic material and attempts have been made to classify, or at least characterize, them according to their genome. However, the known diversity is biased by the techniques implemented for their selection, and by research interests and application. Although advances in bioinformatics tools for the study of metagenomes have allowed a more in-depth analysis of virus diversity [15], the assembly of bacteriophage genomes is still challenging given the modular composition of their genomes. Despite advances in new sequencing techniques and bioinformatics algorithms, little is known about the biology of phages that have been described through genomic sequences. Knowledge about the diversity, ecology, and biology of phages in anaerobic niches is even more limited by the challenges involved in working with obligate anaerobic microorganisms.

The following sections present an overview of the bacteriophage diversity for obligate anaerobes, and the information available on the infection of facultative bacteria in the absence of oxygen. For more detailed information about phage diversity there are several excellent reviews published on the subject [15,16,17].

### 2.1. Phages Against Obligate Anaerobes

The number of phages isolated for strict anaerobic bacteria is small compared to the number of known viruses for aerobic ones such as *Pseudomonas* spp. or facultative anaerobes such as *Escherichia coli* or *Salmonella* sp. Nonetheless, these few phages are important due to the impact of their hosts to the human and animal health.

#### 2.1.1. *Clostridium* spp. Phages

The vast majority of phages reported for obligate anaerobes occur in the genus *Clostridium*, especially in clinically important species such as *C. perfringens* and *C. difficile. C. perfringens* is found in human and animal intestines and is the cause of gas gangrene and necrotic enteritis. Accordingly, bacteriophage isolates for this bacterial species have been prepared for use in pharmacotherapy [18,19]. However, the limited host range of the isolates obtained has diverted interest towards the bacteriophage-related lytic enzymes, which may provide a wider range of action [18]. Given the limited number of isolates (according to ICTV reviewed in august 2020), there are only 3 classified phages against *Clostridium* spp. Information available on complete genomes of phages effective against *C. perfringens* is also limited. In 2012, Morales et al., reported the complete genome sequence for phage ΦCP24R, defining it as virulent for *C. perfringens* as no integrase or lysogenic-related genes were found [20].

*C. difficile*, recently reclassified as *Clostridioides difficile* (together with the species *Clostridioides (Clostridium] mangenotii)* [21], is the main cause of diarrhea associated with antibiotic use. Attempts have also been made to isolate phages for control purposes, but the isolates obtained are of phages with lysogenic lifecycles (Figure 1). These isolates were obtained from environmental and clinical samples, as well as by induction with mitomycin C or UV light [22,23,24]. Until 2018, 24 genomes of *C. difficile* phages were found in public databases, most of them belonging to the family *Myoviridae* [25]. In 2015, Hargreaves and Clokie conducted a taxonomic review comparing 12 genomes of these phages, and attempted to group them by protein notation; they proposed the new genus Phimmp04likevirus (not yet included in the ICTV) [26]. In the absence of strictly lytic phage isolates, studies of *C. difficile* phages have focused on toxin modulation and gene transfer analysis. However, some researchers have proposed its therapeutic use via genetic engineering to modify lysogenic phages able to efficiently decrease populations of this bacterial species. Both in vitro and in vivo trials have been conducted, with promising results for the control of *C. difficile* [27]. It has also been proposed that temperate viruses be used in clinical diagnosis by introducing reporter genes that signal the phage’s active infection process in order to identify the bacteria in the patient’s sample [28].

Phages have also been isolated for the *C. botulinum* and *C. tetani* species. The first reports of bacteriophages for these species date back to the 1960s and, as with *C. difficile*, they refer to the identification of chemically or physically induced prophages [26,29,30,31].

#### 2.1.2. *Bacteroides* spp. Phages

After *Clostridium,* the genus *Bacteroides* probably has the largest number of identified viruses. As species of this genus can be found in human gut microbiota and has been proposed as an indicator of fecal contamination in water. In order, therefore, to develop rapid techniques to identify *Bacteroides* in water, research was conducted in the 1980s and 90s on infectious bacteriophages for *B. fragilis* [32,33,34,35,36,37] but such research soon ceased.

Currently, the research on phages specific for *Bacteroides* is focused on its detection as indicators of human fecal contamination, using molecular techniques [38,39]. Detection of crAss-like phages has received special attention [40,41]. However, until 2018, the first crAss-like phage was isolated and evaluated in vitro: phage ΦCrAss001 infects efficiently *Bacteroides intestinalis* on semi-solid agar and its genome does not present obvious lysogeny genes. However, the infection in liquid culture did not affect the proliferation of the bacteria. The authors suggested that this behavior allows the coexistence of the virus and its host in an equilibrium that could be an ecological advantage in the phage-host interaction within the gut environment [42].

#### 2.1.3. *Desulfovibrio* spp. Phages

Research on bacteriophages of the genus *Desulfovibrio* does not focus on therapy but on ecology; it seeks to understand the dynamics of viruses as nutrient recyclers and influencers of the diversity and abundance of anaerobic bacteria in anoxic niches. Research has focused on this genus as it is the model for sulfate-reducing bacteria, which are widely available in marine and terrestrial environments, and are also well characterized biochemically.

So far, only five phage isolations have been reported for four *Desulfovibrio* species: Handley et al., in 1973, isolated lysogenic phages of *D. vulgaris* induced by *mitomycin C* [43]; Rapp and Wall found phages that could mediate transduction in *D. desulfuricans* [44]; Kamimura and Araki isolated lytic phages that infected *D. salexigens* [45]; Walker et al. also isolated temperate phages for *D. vulgaris* [46]. Finally, Eydal et al. isolated bacteriophages with lytic action effective against *D. aespoeensis* [47]. Since there are no reported genomes for the phages of these bacteria, it is difficult to confirm whether or not they are virulent. The available information on the genome of phages infective for *Desulfovibrio* is restricted to the genome of the model strain *D*. *vulgaris* Hildenborough, for which four prophages have been identified: two phages resembling each other, one lambdoid and the remnant of the genome of a fourth phage [48].

#### 2.1.4. Phages of Other Obligate Anaerobic Species

There are a number of other approaches to the study of obligate anaerobic phages in bacteria that cause problems in the oral cavity of humans. Given the specificity, a common characteristic among most phages, these approaches have focused on the identification of pathogenic bacteria but not on their control as infectious agents. Studies have been conducted on the species *Fusobacterium varium* [49] and *F. nucleatum* [50], and on the genus *Actinomyces* [51,52].

On the other hand, in 2015, Holmes et al. reported the identification of phage sequences in the genomes of five different *Geobacter* species. Some *Geobacter* species are important agents in the bioremediation of organic contaminants and metals, but by some unknown factor, in situ growth of *Geobacter* is limited. The authors suggest that maybe the bacterial host is lysed by induced prophages, when conditions during the bioremediation process are optimal for bacterial growth. This hypothesis is also supported by the finding of genes associated with lysogenesis. They further demonstrated that the transcription and translation of phage genes increases during uranium bioremediation [53].

As is evident, information on phage isolates of obligate anaerobes is scarce, and much of what does exist concerns temperate viruses. The absence of specialized techniques, and the difficulties involved in working with anoxic systems and with microorganisms that are difficult to grow, hinder the search for these bacteriophages. However, at the same time, this opens up great opportunities to create new knowledge.

### 2.2. Phages Against Facultative Anaerobes

The isolation of bacteriophages capable of infecting facultative anaerobes began with their very discovery in 1915. Since then, most research has focused on facultative anaerobic bacteriophages. Of the 3426 bacterial virus genomes reported in the NCBI as of August 2020, more than 1700 correspond to viruses specific for facultative anaerobic bacteria, mostly enteric bacteria.

#### 2.2.1. Gram Negative Enteric Bacteria Phages

Most phages have been isolated against *Escherichia coli*, *Klebsiella pneumoniae*, *Salmonella enterica* and *Shigella* sp. Table 1 presents the number of taxonomically classified bacteriophages for some of the major enteric bacterial genera, according to the *International Committee on Taxonomy of Viruses*—ICTV.

Bacteria of the *Enterobacteriaceae* family are Gram-negative bacilli that cause various diseases such as bacteremia, septic arthritis, endocarditis, respiratory tract infections, skin infections, urinary tract infections, intra-abdominal infections, and ophthalmic infections in humans and animals [54]. As such, given their clinical importance, there is great interest in finding alternatives to control these pathogens.

*E. coli* is certainly one of the most commonly used species for bacteriophage isolation. Bacteriophages T4, Lambda, and M13 are the models for lytic, lysogenic and continuous development lifecycles, respectively, and have served to understand the biology of bacteriophages and elucidate mechanisms of molecular biology. *E. coli* has not only been repeatedly used as a host because it is one of the best known organism in terms of its genome and metabolism but also because it is of clinical importance, especially because of its association with acute diarrhea in humans and animals [55].

On the other hand, the presence of prophages in *E. coli* O157 strains has important implications given the acquisition of fundamental virulence factors for the development of infection. The production of shiga-toxins by *stx* genes is the main example among the virulence factors contributed by prophages in O157 strains [56]. Many of these prophages can be defective, but even these remnants of prophages can be determinants in the pathogenicity of the host [57]. Effector proteins or other genes that expand the metabolic repertoire of the bacterium are also found within the sequences of the prophages. An example of this, and related to the oxygen metabolism, is the *sodC* gene encoding of copper-zinc superoxide dismutase (Cu, ZnSOD), an important component in the defense against reactive oxygen species in aerobic environments. *E. coli* O157:H7 has two additional copies of this gene within two profuse lambdoid phages in its genome. It has been shown that these copies, unlike the bacterial copy, are expressed in the logarithmic phase and not only in the stationary phase, and that their production is not affected by the presence of oxygen in the environment [58].

Both *E. coli*, *K. pneumoniae* and *Shigella* have been studied in animals as models for the treatment of human infections, and in all cases, the results reveal that the infection is controlled following the use of bacteriophages by oral, intravenous or respiratory pathways. In the case of *Salmonella*, the greatest efforts have been made to control the pathogen in birds and pigs [54]. Clavijo et al. reported the use of *Salmonella* bacteriophages in a broiler production system, in which they demonstrated the efficiency of a cocktail of 6 bacteriophages in reducing the presence of the pathogen on the farm [59].

Other studies have advanced in isolating bacteriophages for other enteric bacteria, including *Serratia marcescens*, *Edwardsiella* sp., *Erwinia* sp. and *Citrobacter* sp. [54].

#### 2.2.2. Phages Against Vibrios

Given their importance to human and animal health, numerous bacteriophages have also been isolated for *Vibrio* spp. even though they are not enteric bacteria.

The most interesting species in the genus *Vibrio* is *V. cholerae*, whose bacteriophages have been known since the time of d’Herelle. Studies were first conducted in humans and then moved to animal models because of ethical considerations and the difficulties of having controls on treatments involving research in humans. Studies involving *V. cholerae* bacteriophages are still ongoing, and various investigations have shown that bacteriophages can be effective as a prophylactic or as a treatment. However, results vary depending on the strain and route for phage application [60], where phage treatments can show good protection [61,62] or partial protection [63] against *V. cholerae*. Other human pathogenic *Vibrio* species for which bacteriophages have been isolated are *V. parahaemolyticus, V. vulnificus*, *V. fluvialis*, and *V. furnissii*. Trials have been advanced both to treat the infection of the pathogen and to reduce the burden of the microorganism on foodstuffs that transmit it. In the case of pathogenic vibrios in animals, the isolation of bacteriophages has mainly focused on *V. harveyi* because of its economic importance in aquaculture. Bacteriophage isolates are also available for other species such as *V. anguillarum*, *V. ordalii*, *V. splendidus*, *V. coralliilyticus*, and *V. cyclitrophicus* [60].

#### 2.2.3. Phages Against Gram-Positive Bacteria

Due to their importance in human and animal health, phages have been isolated from Gram-positive bacteria such as *Streptococcus sp*., *Enterococcus faecalis* and *Staphylococcus aureus* [1]. Within the Gram-positive group, *S. aureus* is the bacterium with the most bacteriophage isolations. Many of the *S. aureus* bacteriophages aim to control mainly MRSA-resistant methicillin strains, because of the challenge involved in treating these strains with antibiotics. There are more than 600 reported genomes of *S. aureus* bacteriophages, among which over 200 are lytic viruses [64]. The lytic lifecycle bacteriophage model for this bacterial species is the K bacteriophage, which belongs to the family *Herelleviridae*, is one of the best characterized, and has similarities to several isolates obtained subsequently [65]. The most commonly-known *S. aureus* bacteriophages, including ones with both lytic and lysogenic lifecycles, belong to the family *Siphoviridae*. Almost all reported isolates of virulent bacteriophages specific for *S. aureus* are from the families *Myoviridae* and *Podoviridae*. Likewise, the sequencing of several *S. aureus* strains has shown that all of them have between one and four prophages that encode several virulence factors, meaning that their study is also of great interest at clinical level [66]. Finally, different in vivo trials have shown the efficacy of phage therapy using phages virulent against *S. aureus* and which represent the most popular group among phages characterized for therapeutic purposes [65].

Another facultative anaerobic Gram-positive genus with several bacteriophage isolates is *Bacillus*. In this case, the species *B. cereus* has been the most commonly used to isolate bacteriophages, in order to control them in food. *B. cereus* virus isolates belong to the order *Caudovirales* and to the family *Tectiviridae*; there are no reports of filamentous bacteriophages, single-stranded DNA or RNA [67]. In the case of the genus *Bacillus*, bacteriophages have not only been isolated to control the pathogen but also to typify highly virulent strains, especially in the case of *B. anthracis* [67]. Other facultative anaerobic species of these Gram-positive bacilli that have been used as hosts include *B. mycoides* and *B. pseudomycoides*, although with few reports compared to the other previously mentioned species [68].

Lactic acid bacteria are of vital importance in the food industry in the production of fermented products, with Gram-positive bacteria of the genus *Lactobacillus* being the most commonly used in such processes. However, virulent bacteriophages present a disadvantage in the production processes, as they reduce the bacterial population needed for fermentation. Up until 2009, more than 231 *Lactobacillus* bacteriophages had been reported and identified [69], and more than half belonged to the family *Siphoviridae*. The remaining bacteriophages have been classified into the families *Myoviridae* and *Podoviridae* [70]. *Lactobacillus* bacteriophages are of great interest, not only due to their role in affecting the fermented products industry but because they are also believed to play a key role in the balance of vaginal microbiota by regulating the population of lactic acid bacteria. Indeed, it has been observed that the action of bacteriophages in bacterial vaginosis lowers the *Lactobacillus* population and increases the anaerobic Gram-negative bacilli, and such changes appear to occur due to the induction of prophages, caused by chemical phenomena in the vagina. Indeed, the presence of lysogens in *Lactobacillus* species is very frequent [70].

Another area of interest for bacteriophages isolation to control different pathogens affecting production is in the field of aquaculture. These pathogens include facultative anaerobic Gram-positive bacteria such as *Bacillus licheniformis* [71], *Lactococcus* spp. and *Streptococcus iniae* [72]. Studies on viruses of these species involve phage therapy on fish and shellfish. However, considering the high number of pathogens in fish and the wide range of fish species used in aquaculture, the small number of viruses isolated for these pathogenic bacteria shows that research is still limited in this field.

In sum, and based on the information presented, it is clear that more information is available on the viruses of facultative anaerobes than those of obligate anaerobes. However, as discussed below, little is known about the behavior of the viruses of facultative anaerobes under conditions of oxygen deprivation.

## 3. Infection

Using models such as *Escherichia coli*, *Pseudomonas aeruginosa*, *Bacillus subtilis* or *Staphylococcus aureus*, it has been possible to show that there are differences in the phage infection mechanisms for Gram-positive and Gram-negative bacteria, mainly due to the composition of their walls [73,74,75,76,77,78,79,80,81,82]. The type of receptor and the way in which the genetic material is injected are aspects that change according to the composition of the cell wall and these have been very well described. In contrast, little is known about the environmental parameters associated with successful viral infection. For example, the effect of changes in the bacterial metabolism on viral infection and replication is as yet unknown. Given the close relationship of the phage to its host, changes associated with energy availability or other metabolic processes are likely to affect plating efficiency, progeny size, or latency time. Understanding the effects of environmentally induced metabolic changes in bacteria on phage infection is useful information for phage therapy.

Given the multitude of anaerobic niches in the environment and in living organisms, to explore the effects of changes in oxygen availability on the dynamics of phage-host infection is highly relevant. The transition from aerobic to anaerobic environments probably implies considerable metabolic stress for the bacterial cells, which could affect phage infection efficiency. However, little is known about the effect of oxygen deprivation on phage infection. Below is a review of some infection factors that may be affected by a lack of oxygen, that lead to successful phage therapy, and that are essential in understanding the virus-host interaction.

### 3.1. Infection in Obligate Anaerobes

#### 3.1.1. *Clostridium* spp.

The information available on the characterization of the infection cycle of obligate anaerobes is restricted to temperate *Clostridium difficile* bacteriophages (today *Clostridioides difficile* and referred to here as *C. difficile*). Goh et al. classified four isolates of temperate bacteriophages into two groups: a first group with a short latency period (32–36 min) and small burst size (5–7), and a second group with a long latency period (90–118 min) and large burst size (19–33) [83]. It has also been shown that *C. difficile* temperate bacteriophages have two latency periods: a short first period with little viral progeny and a long one, which they refer to as the complete replication cycle. In this case, the burst size for 7 bacteriophages ranged between 70–360, with a latency period of 60–100 min [28]. The observation concerning the two latency periods is due to the methodology used to estimate the viral progeny and the latency period, as these parameters were evaluated from modifications of an infection curve and not a one-step curve (probably due to the adjustments made to the techniques to avoid oxygen contamination). Regarding the host range, *C. difficile* temperate bacteriophages do not seem to be very specific and most of them are able to infect several ribotypes [28,83,84,85].

No information is available on lytic lifecycle bacteriophages for *C. difficile* because no virulent bacteriophages have been isolated for this bacterial species. However, studies have been conducted on the effect that prophages may have on their host, revealing that *C. difficile* prophages affect the regulation of virulence factors (such as toxin production), an effect that depends on the specific bacteriophage. In the case of the bacteriophage ΦCD119, the results indicate that the production of toxin A is reduced by approximately 50% in the lysogen. From the transcriptional analysis, the expression of the pathogenicity locus, characteristic of *C. difficile*, was observed to decrease in the lysogen, suggesting that prophage induces downregulation of these transcripts, apparently by the repressive action of the RepR protein [86]. In contrast, phage ΦCD38-2 increases the expression levels in the pathogenicity locus during lysogeny, thus increasing the synthesis of the A and B toxins [87].

In the case of *C. perfringens*, there is evidence that prophages in bacterial genomes increase sporulation, making it more resistant to high temperatures [88,89]. In the same line, Sekulovic, in 2015, conducted a global transcriptional analysis of *C. difficile* lysogens with ΦCD38-2, with the results indicating that 39 genes were differentially expressed in the lysogen. Twenty-six of the 39 were downregulated, including transcriptional regulators and genes related to sugar and carbon metabolism, especially phosphotransfer systems. Some of the few genes that were upregulated included genes associated with phosphotransferase systems, but specifically with glucose metabolism. Similarly, the coding gene for the CwpV protein, related to host colonization and normally expressed when controlling for the expression of virulence factors in bacteria, was the most upregulated in the lysogen. Upregulation was also identified for prophages that encode CRISPR system genes, suggesting that this prophage protects the bacterium from external genetic material [85].

#### 3.1.2. *Desulfovibrio* spp.

So far, there have been no reports of viruses infecting *Desulfovibrio*, thus the lifecycle of the known phages for this genus is not known. Some reports in the literature refer to lytic viruses because lysis is observed; however, there is no evidence of whether they can or cannot perform a lysogenic cycle. Despite this, adapting the traditional techniques of bacteriophage manipulation and study, in the case of a *D. aespoeensis* isolate, it has been estimated that the burst size can be close to 170 and with a latency period of 70 h. These values were calculated by adapting the most likely number methodology to estimate the number of viable viruses [47]. Finally, in the case of a *D. salexingens* bacteriophage, burst size was not estimated, but viral titers of 10^9^–10^11^ PFU/mL were obtained, indicating a high viral production per infected cell [45].

According to the above, there are many information gaps regarding bacteriophage infection in obligate anaerobes, especially on virulent bacteriophages. In the case of archaeal viruses, this is no different. Given the strict growth requirements of these organisms, it is even more complicated to use traditional techniques to study the infection and replication cycle. Sometimes, it is not even possible to obtain isolates using the double layer agar technique, and thus electron transmission microscopy is required to purify the isolates, as this makes it possible to identify how many different viral morphologies there may be. Even so, a greater variety of viruses that cause infection in the absence of oxygen are known among Archaea, although the information available about these viruses is limited to the morphology of their virion and the organization of their genome.

### 3.2. Infection in Facultative Anaerobes

As mentioned before, some facultative anaerobes such as *S. aureus* and *E. coli* have been used as host models to understand bacteriophage infection and replication. This knowledge about bacteriophage lifecycles and interactions with their host has been obtained from studies conducted under aerobic conditions, or without considering the oxygen concentration in the environment. Furthermore, no information is available on the differences in bacteriophage infection from facultative anaerobes in aerobiosis and anaerobiosis. Accordingly, this section summarizes the available information on the effect of the infection on the metabolism of facultative anaerobes, in order to identify critical aspects where the absence of oxygen could be crucial for the virus’ infection process.

Bacteriophage infection is usually accompanied by a rearrangement in bacterial gene expression and changes in cell physiology and metabolism [90]. Bacteriophages modify the host’s cellular machinery in order to take control of certain functions. An example of this is T4, which modifies the specificity of host RNA polymerase using two factors, MotA and AsiA that interact with the σ^70^ subunit of *E. coli* RNA polymerase. With these modifications, T4 is able to obtain a preferential transcription of the genes of its genome [91].

Several studies have shown evidence of changes in gene expression and cell physiology during infection. In the infection of *E. coli* bacteriophage PDR1, using microarrays, it was observed that most of the changes in host gene expression took place only following synthesis of the virion components. This suggests that there is no major host reprogramming during the early phase of the infection and no dramatic changes in gene expression in the bacterium. Most of the genes with high-level induction, encode for chaperones and proteins related to induced stress, and were expressed mostly in the late phase of the infection. Throughout this study, when comparing gene expression in bacteria at 5, 10, 15, 30, and 50 min post-infection, no changes were detected in genes involved in cell growth and division. This explains the lack of variation in the rate of replication. Although no dramatic changes were evident in the early phase of the infection, genes involved in osmotic adaptation, anaerobic respiration, organic compound consumption, and cation export were differentially expressed. The authors propose that these changes in expression may be aimed at counteracting the effect of the presence of viral DNA in the bacterial cytoplasm, since it increases the permeability of the membranes, causing a loss of cytoplasmic compounds. Also, the presence of—sometimes single stranded—viral DNA activates the bacterial SOS response system. Finally, sometime after infection, exopolysaccharide synthesis was induced, probably as a mechanism of superinfection immunity [92].

Using RNA-seq, in 2016, Leskinen evaluated how the phage-host interaction changes during an infection cycle in aerobically cultured *Y. enterocolitica*. The results showed that host transcripts decrease over time, apparently as a consequence of bacterial RNA degradation. Immediately after infection, host gene transcription decreased by approximately 2.58%. Furthermore, as in PRD1 with *E. coli* [92], there are not many changes in the bacteria’s gene expression. Among the few differentially expressed genes for *Y. enterocolitica*, seven—related to membrane proteins, involved in energy and electron transport—upregulated. On the other hand, downregulated genes include nitrate reductase and many others in general without any particular trend. In this case, genes also upregulated as a consequence of osmotic stress, probably as a result of changes in the metabolic environment, and upregulated in the late phase of the infection [93].

In 2018, the effect of the infection on the expression profile of the bacterium cultured under microaerophilic conditions was evaluated by means of transcriptomic analysis of a T4-type bacteriophage in *C. jejuni*. In this case, in all assessed time periods, there were more upregulated genes than downregulated ones in response to the infection. Genes related to ribosome pathways were identified among the upregulated ones, and genes related to energy pathways, such as pyruvate metabolism and the tricarboxylic acid cycle, were identified as downregulated ones. This indicates that the metabolism of the bacterium decreases with infection. As in the previous cases, stress-related genes were upregulated, especially oxidative stress pathway genes. The deletion of these genes in the bacteria decreased plaque efficiency, inferring that infectivity can be altered by multiple types of oxidative stress [94].

The above studies focused on virulent bacteriophages. But how can a lysogen change as a result of prophage induction? In 2015, a comparison was made of the transcriptome of *E. coli* strains infected with Stx bacteriophages, before and after triggering the SOS response that initiates the lytic cycle. According to the results obtained by the researchers, downregulation occurs in the lysogen enzymes associated with acetyl-CoA and pyruvate synthesis, and cytochrome-b-oxidase reduction, related to aerobic respiration. Interestingly, all affected pathways are related to energy production. In addition, when prophage induction occurs, the expression of anaerobic respiration genes is reduced and the expression of carbohydrate metabolism, iron acquisition and phosphate metabolism varies [95].

On the other hand, not only do phages lead to changes in bacterial metabolism to ensure successful infection, as a result of the infection, they can also produce changes in cellular physiology to ensure successful replication and prevent superinfection. These phenomena have been described in bacteriophages such as ΦX174 [96] and P1 [97] in *E. coli*.

Based on the above, it is possible to conclude that the development of the viral replication cycle affects the expression of host genes, which in turn causes changes in bacterial physiology and metabolism. It seems that when infection occurs, the metabolic pathways related to carbohydrates and energy generation are reduced, perhaps aiming to diminish bacterial metabolism. Furthermore, although in some cases the bacteria were not cultured by limiting the oxygen in the medium, surprisingly, the pathways related to anaerobic respiration are also affected either positively or negatively in terms of expression.

Finally, if bacteriophage infection affects the expression of aerobic and anaerobic energy generation pathways, added to the fact that in the absence of oxygen the amount of energy produced by the cell is lower, and that the genes related to anaerobic respiration will be upregulated to favor bacterial survival, it would be interesting to consider how facultative anaerobic phage infection is affected in anaerobic environments. It would therefore be plausible to assume that the bacteriophage must be prepared to respond to these environmental changes. Perhaps some phage genes that cannot normally be annotated and are proposed as hypothetical proteins, take on a fundamental role in anoxic systems, and have not been described so far since there are no studies on infection in anaerobic conditions. In unpublished data, when comparing the infection and replication cycle of *Salmonella* phage ϕSan23 in aerobic and anaerobic conditions, it was found that in the absence of oxygen, infection efficiency is reduced as is burst size [98], and the appearance of bacterial clones resistant to the bacteriophage is low (Figure 2). This suggests that the presence or absence of oxygen does indeed affect viral infection and replication. However, as already mentioned, there is no information available on the infection processes in anaerobiosis, and thus the molecular mechanisms involved during virus infection in the absence of oxygen are also unknown.

## 4. Considerations for Facultative Anaerobic Phage Infection in the Absence of Oxygen

Infection efficiency can be affected by different parameters such as host replication rate, multiplicity of infection, environmental conditions (agitation, temperature, amount of nutrients) or by the host’s physical conditions such as availability of cell surface receptors, which can vary by low levels of expression in response to certain stimuli.

Although the genus *Mycobacterium* is defined as aerobic, some species are able to resist hypoxia. In 2014, using oxygen-limiting mycobacteria, Swift et al., observed that phage TM4 was able to infect the bacteria in hypoxia, while phage D29 could not. Their study also revealed that D29 was capable of binding to the cell surface, even though it could not complete the lytic cycle. The results of the study indicate that under oxygen-limiting conditions, the bacterial cells enter a growth stage that prevents D29 infection, but that does not affect TM4 infection [99].

In *Salmonella* sp. one of the receptors recognized by bacteriophages is the BtuB protein [100], present in the external membrane of certain enteric bacteria among which *E.coli* and *Salmonella* sp., and necessary for efficient vitamin B12 transportation to the cell interior. A high concentration of Ado-B12 inside the cell (one of the forms of B12) is known to repress *btuB* gene encoding [101,102]. For *E. coli*, it has been observed that when the bacterium is cultured with vitamin B12 in the medium, the amount of BtuB on the outer membrane decreases, it is believed, as a result of the increased concentration of the vitamin inside the cytoplasm [103,104]. *Salmonella* sp., on the other hand, is known to be able to synthesize vitamin B12 de novo only in anaerobiosis [105]. Thus, if the regulation mechanisms for BtuB synthesis are similar among enteric bacteria, *Salmonella* sp. may have less BtuB protein in its outer membrane under anaerobic conditions given that it does not require the external supply of the vitamin, as it can synthesize it. Thus, infection efficiency could be affected by the lack of receptors required for host recognition.

The transition from aerobic to anaerobic metabolism in facultative bacteria mainly involves modifications in cell respiration and carbon metabolism, which directly affect energy generation. Metabolic regulation of anaerobiosis in facultative bacteria is given by the ArcA and FNR, in response to oxygen limitation. In *Salmonella* Typhimurium, ArcA and FNR repress pathways associated with aerobic metabolism, energy generation, amino acid and fatty acid transport, and activate gene expression involved with anaerobic metabolism, flagellar biosynthesis, mobility and sugar transportation [106,107]. In *E. coli* anaerobiosis, also associated to the ArcA and FNR systems, the citric acid cycle is transformed to a non-cyclic form and many of the substrates are converted to acetate, ethanol, hydrogen, and carbon dioxide [108]. The above is a demonstration of how anaerobic metabolism directly affects energy generation. Some studies have also focused on the assessment of ATP production under aerobic and anaerobic conditions: in *Aerobacter aerogenes*, it is estimated that in anaerobic conditions ATP production decreases from 36 ATP molecules to three using nitrate as an electron acceptor [109], and in *Pseudomonas stutzeri*, ATP production only occurs in low oxidative phosphorylation from three ATP per oxygen atom to two ATP per reduced nitrate [110].

As phages use the cellular machinery of their hosts for genetic material and protein synthesis, and for the formation of virions, which all require energy, reduced energy production in anaerobiosis may be a determining factor in the development of phage infection. However, it has also been identified that during the infection process, the transcripts and metabolites associated with energy generation pathways are suppressed [94,111,112]. Phages are then likely to benefit from the metabolic changes in the bacterium when there is a delay in cell division and take advantage of existing resources in the cell to form sufficient viral progeny. In fact, De Smet et al., in a comparative analysis of six phages capable of infecting *P. aeruginosa*, found that metabolites such as amino sugars and sugar nucleotides (related to peptidoglycan and lipopolysaccharide synthesis, required for cell growth and division) accumulate in the bacterial cell during infection. The accumulation of these metabolites may be an indication of how the phage inhibits cell division by preventing cell wall and membrane synthesis [112].

## 5. Conclusions

Knowledge of phage-host interaction has been largely focused on aerobic systems, so there is an information gap concerning the possible effect of oxygen-limitation on the infection mechanisms, viral cycles or other parameters associated with phage infection. Many of the niches where phages interact with their bacterial hosts are anaerobic and some of these can be important in phage therapy; for example, the intestines of animals or some types of wounds where there is little or no oxygen available, which is why it is so important to understand virus-host interaction in environments where oxygen is limited.

The results obtained from the study of phage-host interaction in the absence of oxygen are not only interesting in terms of the applicability of phage therapy but may also contribute new information. For example, understanding the function of some of the ORFs and proteins encoded within phage genomes, which have no known function and whose action could be associated with certain hitherto unexplored conditions such as anaerobiosis.

Finally, tests on facultative anaerobes suggest that where oxygen limitation affects the cells metabolism, this in turn, affects successful phage replication. Similarly, transcriptome studies have provided evidence that aerobic and anaerobic metabolism and oxygen reduction mechanisms play a role in the virus infection and replication cycles. As such, energy generation, physiological changes induced in bacterial cells, and host metabolism alterations are aspects to be taken into account in future research on phage infection in the absence of oxygen.

## Figures and Tables

**Figure 1 viruses-12-01091-f001:**
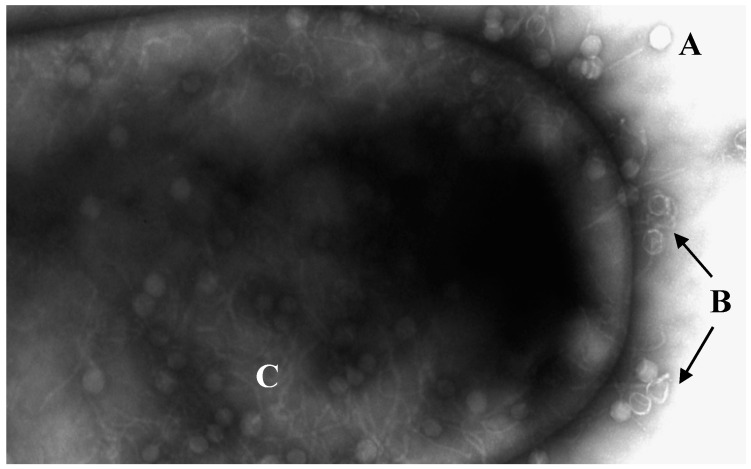
Electronic micrograph of *C. difficile* being infected by bacteriophages. (**A**) Phages without DNA injection. (**B**) Empty capsules by injection of DNA. (**C**) Viral replication inside the bacterium. Transmission electron microscopy performed at the University of Leicester Core Biotechnology Services Electron Microscopy Facility.

**Figure 2 viruses-12-01091-f002:**
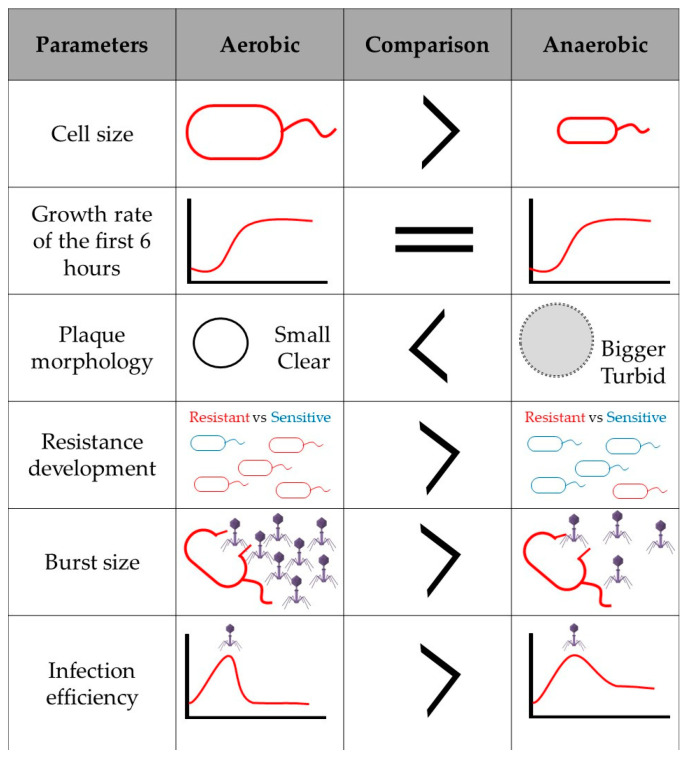
Infection and viral replication of *Salmonella* s25pp bacteriophage ϕSan23 in aerobiosis and anaerobiosis.

**Table 1 viruses-12-01091-t001:** Number of bacteriophages classified by ICTV for some Gram negative enteric bacterial genera. Information taken from Virus Metadata Repository: May 1, 2020 version; MSL35 ICTV.

Host Genus	Number of Bacteriophages (Total Bacteriophages in Database: 2070)	Genetic Material			*Caudovirales: Myoviridae*	*Caudovirales: Podoviridae*	*Caudovirales: Siphoviridae*	Other Families
		ssRNA	ssDNA	dsDNA				
*Escherichia*	276	4	17	255	87	29	32	128
*Salmonella*	109	−	1	108	32	8	24	45
*Klebsiella*	84	−	−	84	15	1	−	68
*Shigella*	36	−	−	36	13	5	1	17
*Erwinia*	31	−	−	31	20	2	2	7
*Aeromonas*	20	−	−	20	11	−	1	8
*Yersinia*	25	−	−	25	6	−	−	19
*Citrobacter*	16	−	−	16	5	−	−	11
*Enterobacter*	12	−	−	12	2	−	−	10
*Edwardsiella*	6	−	−	6	4	1	1	−
*Serratia*	6	−	−	6	2	−	×	4

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
