# Peer review of "Phages in Anaerobic Systems"

_viruses, 2020, doi:10.3390/v12101091_

Round 1
Reviewer 1 Report
This review manuscript by Hernández and Vives entitled “Phages in anaerobic systems” provides an extensive insights on bacteriophages active on obligate anaerobes and discusses how anaerobiosis can be an important parameter in phage infection, especially among facultative anaerobes.
I appreciate the authors clarity of the manuscript, organization of the paper and figures. There are only a couple minor spelling mistakes.
Line 339: change UFP to PFU
Figure 2: change Comparation (on the column) to Comparsion
Author Response
Dear Reviewer,
We thank you for the comments and help to improve the manuscript.
Accordingly, we changed:
- typo in name of Dr. Symour Benzer in line 56.
- verb have was changed for has in line 116.
- line 339 (now line 346) UFP to PFU.
- Archea for Archaea in line 354.
- figure 2, Comparation for Comparison.
- Typos in the reference section.
Best regards,
Santiago Hernández and Martha Vives

Reviewer 2 Report
The authors presented a comprehensive review. It would be great if they could:
- Shorten the text, as some of the general historical background information redundantly appeared in several places.
- Give sufficient references in several places, such as lines 47- 74, 76-79, 93-95, 111-112, 275-277.
- Check the typos, such as line 57, Dr. Seymour Benzer's name was misspelled.
Author Response
Dear Reviewer,
We thank you for the comments and help to improve the manuscript.
Accordingly, changes were performed as follows:
- We shortened the text in
- lines 158-160 "Twort, in 1915, observed lysis plates in cultures of Staphylococcus aureus. d'Herelle, as co-discoverer and after observing lysis, in 1917, used Shigella bacteriophages to control dysentery in soldiers" were eliminated.
- lines 358-361 "E. coli bacteriophages have been the most widely used biological models in history; T4 is the lytic lifecycle model, Lambda is the lysogenic lifecycle model, M13 is the continuous lifecycle model, and ΦX174 is the rolling circle replication model for single-stranded DNA viruses" were eliminated.
- We added the following references:
- Line 62:
Clokie, M.R.J.; Kropinski, A.M. Bacteriophages: Methods and Protocols, Volume 1: Isolation, Characterization, and Interactions; Humana Press: 2009.
- Line 67:
Dion, M.B.; Oechslin, F.; Moineau, S. Phage diversity, genomics and phylogeny. Nat Rev Microbiol 2020, 18, 125-138, doi:10.1038/s41579-019-0311-5.
- Line 97:
Phothichaisri, W.; Ounjai, P.; Phetruen, T.; Janvilisri, T.; Khunrae, P.; Singhakaew, S.; Wangroongsarb, P.; Chankhamhaengdecha, S. Characterization of Bacteriophages Infecting Clinical Isolates of Clostridium difficile. Front Microbiol 2018, 9, 1701, doi:10.3389/fmicb.2018.01701.
Hargreaves, K.R.; Clokie, M.R. Clostridium difficile phages: still difficult? Front Microbiol 2014, 5, 184, doi:10.3389/fmicb.2014.00184.
Seal, B.S.; Oakley, B.B.; Morales, C.A.; Svetoch, E.A.; Siragusa, G.R.; Garrish, J.K.; Simmons, M.; Volozhantsev, N.V. Bacteriophages of Clostridium perfringens; INTECH Open Access Publisher: 2012
- Line 120 (previously, 111-112):
Tartera, C.; Lucena, F.; Jofre, J. Human origin of Bacteroides fragilis bacteriophages present in the environment. Appl Environ Microbiol 1989, 55, 2696-2701, doi:10.1128/AEM.55.10.2696-2701.1989.
Jofre, J.; Blanch, A.R.; Lucena, F.; Muniesa, M. Bacteriophages infecting Bacteroides as a marker for microbial source tracking. Water Res 2014, 55, 1-11, doi:10.1016/j.watres.2014.02.006.
Booth, S.J.; Van Tassell, R.L.; Johnson, J.L.; Wilkins, T.D. Bacteriophages of Bacteroides. Rev Infect Dis 1979, 1, 325-336, doi:10.1093/clinids/1.2.325.
- Line 282 (previously, 275-277):
Rakhuba, D.V.; Kolomiets, E.I.; Dey, E.S.; Novik, G.I. Bacteriophage receptors, mechanisms of phage adsorption and penetration into host cell. Pol J Microbiol 2010, 59, 145-155.
Young, R. Bacteriophage lysis: mechanism and regulation. Microbiol Rev 1992, 56, 430-481.
Dunne, M.; Hupfeld, M.; Klumpp, J.; Loessner, M.J. Molecular Basis of Bacterial Host Interactions by Gram-Positive Targeting Bacteriophages. Viruses 2018, 10, doi:10.3390/v10080397.
Li, X.; Gerlach, D.; Du, X.; Larsen, J.; Stegger, M.; Kühner, P.; Peschel, A.; Xia, G.; Winstel, V. An accessory wall teichoic acid glycosyltransferase protects Staphylococcus aureus from the lytic activity of Podoviridae. Sci Rep 2015, 5, 17219, doi:10.1038/srep17219.
Daugelavicius, R.; Cvirkaite, V.; Gaidelyte, A.; Bakiene, E.; Gabrenaite-Verkhovskaya, R.; Bamford, D.H. Penetration of enveloped double-stranded RNA bacteriophages phi13 and phi6 into Pseudomonas syringae cells. J Virol 2005, 79, 5017-5026, doi:10.1128/JVI.79.8.5017-5026.2005.
Baptista, C.; Santos, M.A.; São-José, C. Phage SPP1 reversible adsorption to Bacillus subtilis cell wall teichoic acids accelerates virus recognition of membrane receptor YueB. J Bacteriol 2008, 190, 4989-4996, doi:10.1128/JB.00349-08.
Vinga, I.; Baptista, C.; Auzat, I.; Petipas, I.; Lurz, R.; Tavares, P.; Santos, M.A.; São-José, C. Role of bacteriophage SPP1 tail spike protein gp21 on host cell receptor binding and trigger of phage DNA ejection. Mol Microbiol 2012, 83, 289-303, doi:10.1111/j.1365-2958.2011.07931.x.
Ghose, C.; Euler, C.W. Gram-Negative Bacterial Lysins. Antibiotics (Basel) 2020, 9, doi:10.3390/antibiotics9020074.
Zagotta, M.T.; Wilson, D.B. Oligomerization of the bacteriophage lambda S protein in the inner membrane of Escherichia coli. J Bacteriol 1990, 172, 912-921, doi:10.1128/jb.172.2.912-921.1990.
Bläsi, U.; Henrich, B.; Lubitz, W. Lysis of Escherichia coli by cloned phi X174 gene E depends on its expression. J Gen Microbiol 1985, 131, 1107-1114, doi:10.1099/00221287-131-5-1107.
- We corrected
- typo in name of Dr. Seymour Benzer in line 56.
- verb have was changed for has in line 116.
- UFP for PFU in line 346.
- Archea for Archaea in line 354
- figure 2, Comparation for Comparison.
- Typos in the reference section.
Best regards,
Santiago Hernández and Martha Vives
